# PyTupli: A Scalable Infrastructure for Collaborative Offline Reinforcement Learning Projects

## Abstract

Offline reinforcement learning (RL) has gained traction as a powerful paradigm for learning control policies from pre-collected data, eliminating the need for costly or risky online interactions. While many open-source libraries offer robust implementations of offline RL algorithms, they all rely on datasets composed of experience tuples consisting of state, action, next state, and reward. Managing, curating, and distributing such datasets requires suitable infrastructure. Although static datasets exist for established benchmark problems, no standardized or scalable solution supports developing and sharing datasets for novel or user-defined benchmarks. To address this gap, we introduce PyTupli, a Python-based tool to streamline the creation, storage, and dissemination of benchmark environments and their corresponding tuple datasets. PyTupli includes a lightweight client library with defined interfaces for uploading and retrieving benchmarks and data. It supports fine-grained filtering at both the episode and tuple level, allowing researchers to curate high-quality, task-specific datasets. A containerized server component enables production-ready deployment with authentication, access control, and automated certificate provisioning for secure use. By addressing key barriers in dataset infrastructure, PyTupli facilitates more collaborative, reproducible, and scalable offline RL research.

## 1 Introduction

Reinforcement learning (RL) algorithms provide effective solution approaches for decision-making under uncertainty, but require interaction with the real system or a simulation model that can be costly. As the success of machine learning commonly relies on the availability of large amounts of data, offline RL has emerged as a paradigm that decouples RL from the necessity of online interactions (Lange et al., 2012). An offline RL agent is trained on a dataset of tuples *(state, action, next state, reward)* that can, for example, be obtained from historical data. Consequently, publicly available tuple datasets for benchmark problems such as D4RL (Fu et al., 2020) or Minari (Younis et al., 2024) are indispensable in the process of developing more powerful offline RL algorithms (Kumar et al., 2020; Kostrikov et al., 2021). These static dataset collections span several domains such as robotics and games (Fu et al., 2020; Younis et al., 2024; Formanek et al., 2023; Gulcehre et al., 2020), power system control (Qin et al., 2022), and autonomous driving (Liu et al., 2023; Lee et al., 2024), and are often handcrafted to address specific challenges following the question: *"How can we improve existing offline RL algorithms?"*. However, a question that is often much more relevant for RL practitioners is: *"Which offline RL algorithm is the best one for solving my problem?"*.

Any researcher wanting to answer this question has to create a tuple dataset in a format accepted by one of the standard offline RL libraries such as d3rlpy (Seno & Imai, 2022) or CORL (Tarasov et al., 2022). Furthermore, if they want to train on different devices or share the dataset, for example, in the context of an industry project, they have to set up some infrastructure for this. If the offline RL controllers should be tested or finetuned with online interactions, this includes sharing the benchmark problem itself to provide access to the simulation model. Streamlining this process would greatly facilitate collaboration in both research and industry contexts.

To address this gap, we present PyTupli, a Python tool for creating and sharing tuple datasets for custom environments that follow the gymnasium framework (Towers et al., 2024). Through containerization, PyTupli enables users to host their own database with a concise API for uploading, downloading, and sharing benchmarks and the corresponding tuple datasets. Benchmarks are stored as JSON serialized objects with the possibility of storing related artifacts, such as time series data, algorithm hyperparameter configurations, or trained policies, as separate objects that multiple benchmarks can reference. As it is aimed at scalable collaboration, PyTupli also features user access management. Since the success of offline RL often depends on the quality of the dataset, we offer extensive filtering capabilities. While established datasets only allow filtering episodes (Younis et al., 2024; Liu et al., 2023), we also enable filtering for tuples, which can, for example, help balance a dataset with sparse rewards.

In summary, our contributions are

- a wrapper for custom gymnasium environments that enables recording tuples,

- a novel approach for serializing such custom gymnasium environments, including the possibility to store related artifacts,

- a production-ready container stack with an API server for uploading, downloading, and sharing serialized benchmark problems and associated tuple datasets, and

- advanced filtering capabilities for curating custom datasets for offline RL from existing tuples.

The remainder of this work is structured as follows. In Sec. 2, we provide a motivating example along with the functional requirements for our tool. Sec. 3 then introduces the client and server side of the framework. After detailing how the motivating example could be realized with PyTupli in Sec. 4, we shortly discuss the limitations and conclude in Sec. 5.

## 2 PROBLEM STATEMENT

### 2.1 A MOTIVATING EXAMPLE

We use a motivating example to illustrate the gap in existing infrastructure before formulating requirements for a potential solution. Let us consider a research collaboration between University A and Company B. Company B sells energy management systems (EMS) using rule-based algorithms. Many of their customers are private households with a similar system setup: a photovoltaic generator, a battery energy storage system, and an air-to-air heat pump. With the advent of dynamic electricity tariffs for private households, B wants to investigate the potential of offline RL-based controllers for their EMS and, therefore, starts a joint research project with the RL expert team at University A. The idea is to use historical data provided by B to train a baseline agent with offline RL, which can then be finetuned for each individual household using a small number of online interactions.

The project partners A and B need some infrastructure for exchanging the definition of the control task (the gymnasium environment), related time series data (e.g., load and generation profiles), existing (*state, action, next state, reward*) tuples from historic data, and newly generated interactions from the continuous operation of the EMS. Multiple similar benchmark problems have to be generated for the individual households, and the relations between tuples and benchmarks must be preserved. While version control tools such as GitHub provide some of the desired functionalities, they are not designed for large datasets and do not offer support for tracking complex relationships within datasets nor for efficient querying and filtering. Databases are a better fit, but require some experience for designing efficient workflows.

One can imagine several similar use cases where an offline-RL-centered collaboration would benefit from a tool that automates the process of setting up the required infrastructure for creating, sharing and curating datasets. Furthermore, even individual projects would benefit from the data management functionalities of the described infrastructure. Next, we provide a more generalized version of the requirements for such a tool.

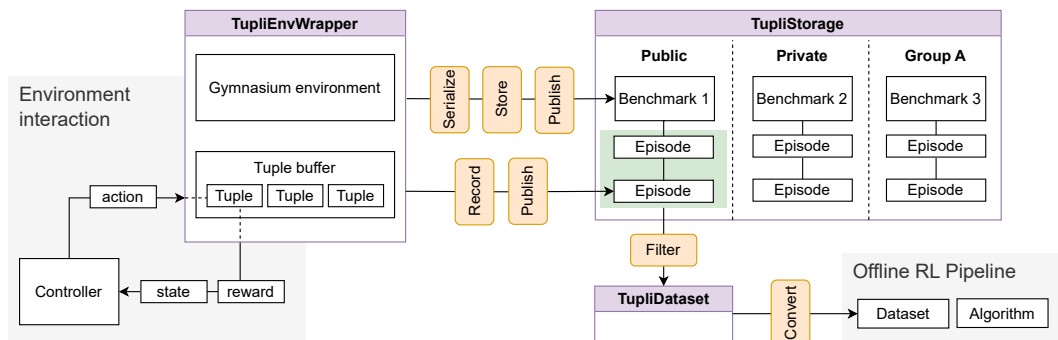

Figure 1: Overview of the core functionalities of PyTupli.

## 2.2 FUNCTIONAL REQUIREMENTS

**R1 Benchmark and Artifact Management** Users need to be able to store any control task representation that is given in the form of a gymnasium environment. An environment may have customizable parameters that lead to small variations in the task definition. We refer to a fully specified environment as a benchmark problem that is unique and can be used to compare the performance of controllers. Benchmark problems may rely on additional data, for example, exogenous inputs used for simulation or pre-trained models. We define such units of data with unknown structure as artifacts which may be referenced by multiple benchmarks. To avoid duplicates, artifacts should be stored separately and only the link to them should be stored in the benchmark problem.

**R2 Data Management** The tool must support ingesting, storing, and querying structured datasets (RL tuples), including their relation to existing benchmark problems and any relevant metadata.

**R3 Multi-User Collaboration and Access Control** Collaboration among multiple users or organizations has to be supported. Based on their role, users can store, retrieve, delete, and publish objects.

**R4 Integration with Existing Offline RL Infrastructure** An interface to the gymnasium framework should enable users to record interactions with gymnasium environments as RL tuples. Furthermore, retrieved tuple datasets have to be made available in a form that can easily be converted into the dataset formats used by existing offline RL libraries such as d3rlpy (Seno & Imai, 2022).

## 3 FRAMEWORK

PyTupli consists of a client-side library that simplifies the workflow in collaborative offline RL projects and a server component that realizes the required infrastructure. Fig. 1 illustrates the core capabilities. After defining a control task using a gymnasium environment, users can invoke PyTupli to convert this task into a unique benchmark problem that can be stored as a serialized object. To store a dataset of RL tuples associated with this benchmark, the user has two options: He can upload a static dataset or record interactions with the environment itself. The benchmark and corresponding dataset can be shared with groups of users or made public, in which case they can be accessed without authentication. Users with appropriate rights can then download and filter this data and use it as the input for an offline RL algorithm. In the following, we provide a more detailed overview of how these functionalities are realized in PyTupli.

## 3.1 CLIENT

The client side of PyTupli has three core classes: `TupliStorage`, `TupliEnvWrapper`, and `TupliDataset`. Fig. 2 illustrates the relations between them. The `TupliEnvWrapper` enables users to create benchmarks from custom gymnasium environments and is described in more detail in Sec. 3.1.1. How to create, retrieve, and curate datasets is explained in Sec. 3.1.2. For

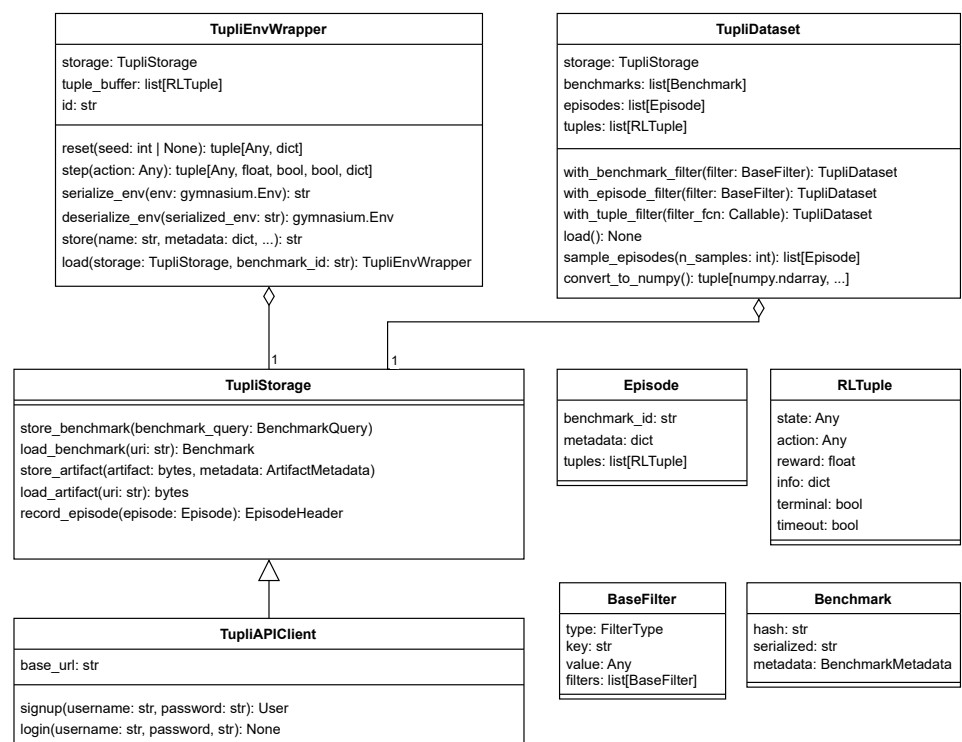

Figure 2: Simplified UML class diagram of the client-side architecture. Some relations are omitted for clarity but can be derived from the given types.

both the `TupliEnvWrapper` and the `TupliDataset`, the `TupliStorage` realizes the interface to the backend. We differentiate between two subclasses of the `TupliStorage` that correspond to two different storage types: The `FileStorage`, which stores objects locally, and the `TupliAPIClient`, which uses MongoDB (MongoDB Inc., 2009) as the storage backend. Both options provide functionalities for storing, retrieving, publishing, listing, and deleting benchmarks and artifacts. For episodes, retrieval is handled via the `TupliDataset`. The `TupliAPIClient` additionally specifies methods for user management, which can also be accessed using a command-line interface (CLI) as described in Sec. 3.1.4. When using the API, users are required to provide authentication for most endpoints, as further explained in Sec. 3.2.1. The `TupliAPIClient` abstracts the required credential management and stores obtained tokens after login securely on the client machine.

### 3.1.1 BENCHMARK CREATION AND STORAGE

PyTupli stores benchmarks as JSON serialized objects with a unique identifier and associated metadata including the benchmark name and a description. The `TupliEnvWrapper`, which inherits from the `Wrapper` class provided by the gymnasium package, serves as a customizable user interface that realizes these functionalities. The `store()` function first serializes the environment, then computes an SHA-256 hash based on the resulting string, and sends a `BenchmarkQuery` containing the hash, the serialized environment, and user-defined metadata to the storage backend. Any change in the gymnasium environment will result in a different hash and thus a new benchmark. The `serialize()` function invokes the `jsonpickle` package to encode the environment. This may not work for all custom environments, but users can overwrite this method to implement their own encoding. As part of the serialization, artifacts such as time series data or pre-trained models can be stored as byte data. The conversion has to be implemented by the user within the `_serialize()` method. It has to invoke the storage backend to store the serialized artifact and its metadata. The backend returns a hash representing the artifact, which is then embedded in the environment in place of the original artifact to enable retrieval during deserialization.

When uploading a benchmark, the backend returns the benchmark id, which is stored internally and can be used to `publish()` or `delete()` the benchmark. Loading the benchmark is a class method of `TupliEnvWrapper` that retrieves the serialized benchmark from the storage using its id and then deserializes it. For any user-defined custom serialization, the corresponding decoding steps have to be specified by overwriting the `deserialize()` or `_deserialize()` methods.

### 3.1.2 DATASET CREATION AND RETRIEVAL

A core objective of PyTupli is storing RL tuples in a structured way and retrieving them as datasets used for offline RL training based on user-defined criteria. To this end, we define two data types, `RLTuple` and `Episode`. An `RLTuple` represents an interaction of a control policy with the environment and is composed of `state`, `action`, `reward`, `info`, `terminated`, and `timeout`. Multiple sequential interactions form an episode, which is ended if either the `terminated` or `timeout` variable is true. An instance of the `Episode` class always has a `benchmark_id` and `tuples`, and can contain additional `metadata`, e.g., whether the episode was generated by an expert policy. Metadata has to be defined by the user and can be used for filtering as described below.

PyTupli supports continuous, incremental data collection for existing benchmarks through two complementary approaches. Static data, e.g., from historical measurements, can be uploaded at any time by using the `record()` functionality of the chosen `TupliStorage`. This requires the user to provide the data in the pre-defined types for `RLTuple` and `Episode`. Users can repeatedly call this functionality to incrementally expand datasets as new data becomes available. As a second option, the `TupliEnvWrapper` enables ongoing data collection by recording all interactions with a gymnasium environment as tuples associated with the respective benchmark. Within the `step()` functionality of the `TupliEnvWrapper`, each interaction is saved in a buffer. When an episode ends, it is sent to the storage automatically, and the buffer is cleared. This allows for seamless accumulation of data over multiple training sessions or experimental runs. Recording episodes can be switched on and off for user convenience.

The primary interface for retrieving tuples is the `TupliDataset` class. When creating an instance of this class, filters can be applied at three different levels using the `with_benchmark_filter()`, `with_episode_filter()`, or `with_tuple_filter()` functionalities. A user could, for example, filter for benchmarks with the same task definition but with varying difficulty by passing the respective benchmark filter. An example for an episode filter is the level of expertise of the behavior policy, which can be specified in the episode metadata. The possibility of filtering tuples could be used to increase the percentage of tuples with high rewards.

When calling the `load()` function of a dataset, the filters are applied in the order benchmark – episodes – tuples. If the dataset is accessed via the API, benchmark and episode filters are executed on the server, while tuple filters are applied locally on the client, using a user-defined callable for maximum flexibility. Sec. 3.1.3 details how to construct complex filters from pre-defined types. The `TupliDataset` additionally provides the option to filter a fixed number of episodes using `sample_episodes()`.

After curating the dataset, it has to be converted into a format supported by the respective offline RL algorithm. Since this varies depending on the library, the `TupliDataset` offers conversions to Numpy arrays and PyTorch and TensorFlow tensors. Furthermore, we provide a converter to the format of the D4RL (Fu et al., 2020) dataset, which is used in many popular libraries such as CORL (Tarasov et al., 2022) or OfflineRL-Kit (Sun, 2023).

### 3.1.3 FILTERING

To define complex dataset queries, PyTupli offers a set of filters implemented as subclasses of `BaseFilter`. Each filter has a `type`, `key`, and `value` field. Atomic filters, such as `FilterEQ` (equals) or `FilterGT` (greater than) apply a simple condition to a key in either the benchmark or episode metadata. These can be composed into logical expressions using `FilterAND` and `FilterOR`, which take a list of filters as input. For convenience, PyTupli overloads the `&` and `|` operators to support the intuitive construction of nested filters using standard Python syntax.

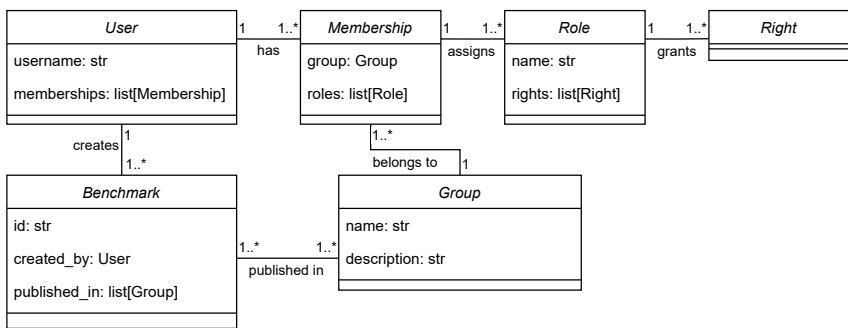

Figure 3: Role-based access management in PyTupli enables fine-grained control for complex collaborative scenarios.

### 3.1.4 COMMAND LINE INTERFACE

To enhance user experience and simplify interaction with stored data, PyTupli includes a lightweight CLI built with the Fire library (David Bieber and Google Inc., 2017). The CLI allows users to directly interact with the backend, such as listing benchmarks, episodes, or artifacts, without writing Python code. It wraps the `TupliAPIClient`, exposing its functionality through intuitive shell commands. Results are displayed in neatly formatted tables, with support for structured output from both Pydantic models and plain dictionaries. This makes the CLI especially convenient for user administration, quick inspection of stored objects, and rapid iteration in development or collaborative settings.

### 3.2 SERVER

The PyTupli server exposes a REST API that allows clients to interact with stored data through a number of endpoints. The endpoints are grouped around the types of objects they manipulate, i.e., benchmarks, artifacts, and episodes. Furthermore, there are endpoints for access and user management. The complete list of endpoints is provided in the Appendix (Tab. 2). The server is implemented using FastAPI (Sebastián Ramírez, 2018), and we use MongoDB (MongoDB Inc., 2009) as a database, as it provides the flexibility to store all considered objects using a single infrastructure component. Most objects are stored directly in a JSON representation, while the GridFS extension (MongoDB Inc., 2009) facilitates file storage for artifacts.

When using endpoints for object creation, the server executes a range of checks to avoid duplication or invalid references. Specifically, for benchmarks, we evaluate if an object with an identical hash already exists that is either owned by the current user or is public. In that case, the operation is rejected. For artifacts, we compute the hash of the content of the artifact concatenated with the id of the current user. If an artifact with this hash already exists, we do not create a new artifact, but no error is thrown. For episodes, we make sure that the referenced benchmark id exists and is either owned by the current user or in a group to which the current user has access.

### 3.2.1 ACCESS & SECURITY

The PyTupli server implements role-based access management. Users can be assigned membership to groups, granting them a set of roles within the group. A role represents a collection of granular rights on a resource, such as create or read. Several common roles, such as contributor or group admin, are predefined. Additionally, entirely custom roles can be created via the server API. A conceptual visualization can be found in Fig. 3, with a full list of predefined roles provided in the Appendix (Tab. 1).

Users can either sign up themselves (OPEN_SIGNUP_MODE = True) or require admins to do so. Each user has a personal group to which their assets are initially associated. Only global admins and the users themselves have access to this private area. Users can then choose to publish an asset in one or several groups. This includes a public group, to which unauthenticated users have read access if the server was configured accordingly (OPEN_ACCESS_MODE = True). To publish in a group, the

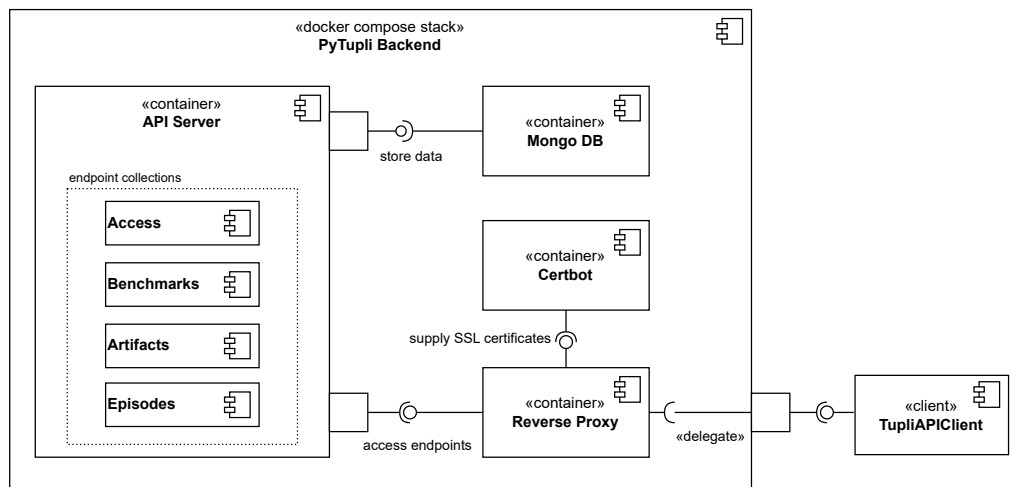

Figure 4: UML component diagram of the production deployment.

user must be a group member with content creation rights, e.g., via the predefined contributor role. Asset owners can, at any time, choose to unpublish their asset again. All users can create groups to which they are automatically assigned admin rights. They can then add and remove members. Other group members can be authorized to manage members via appropriate group update rights. However, they can only ever assign roles the rights of which are a subset of their own rights in the group.

The passwords of users are securely stored using state-of-the-art salted hashing implemented via the bcrypt library (The Python Cryptographic Authority, 2013). Authentication is based on an access token (JSON Web Token) that the client obtains when logging in. For increased security, this access token expires after 60 minutes, after which a refresh token is used to fetch a new access token without the need to provide login credentials again. The refresh token expires after 30 days.

### 3.2.2 Deployment

We provide a Docker Compose (Docker, Inc., 2014) setup for production-ready deployment, as visualized in Fig. 4. Here, the API server is hidden behind an Nginx web server (NGINX, Inc., 2004) acting as a reverse proxy for increased scalability. Further, the reverse proxy takes care of encrypting the communication to the client. Certificate management is automated by running the Let's Encrypt Certbot (Electronic Frontier Foundation, 2015) in a dedicated container. The provided SSL certificates are accepted by all standard browsers and refreshed automatically. By default, a container running a MongoDB database is spawned in the deployment stack. However, users can provide credentials to external infrastructure and exclude the MongoDB container from the stack via a simple flag. For local use, a simplified Docker Compose setup is provided that only includes the API server container and, if enabled, the database container.

## 4 Usage

We use the motivating example introduced in Sec. 2.1 to demonstrate the usage of PyTupli. Let us first consider the workflow of Company B, who has decided to use the `TupliAPIClient` with MongoDB as a backend to realize the infrastructure for this project. After cloning the PyTupli repository, Company B can run the provided docker container stack with default settings using

```
docker compose up --build
```

to start the application, for example, on one of its servers. The subsequent steps are given as a simplified code example in Fig. 5. First, Company B creates the required user accounts, one for its employee Bob and one for the researcher Alice from University A. Then, Bob has to model an exemplary household as a gymnasium environment, which we will not specify here. This system

```
class CustomTupliWrapper(TupliEnvWrapper):
  def _serialize(self, env) -> Env:
    related_data_sources = []
    for ds in env.unwrapped.data_sources:
      metadata = ArtifactMetadata(name=ds.info)
      content = ds.data.to_csv(encoding='utf-8')
      content = content.encode(encoding='utf-8')
      ds_metadata = self.storage.store_artifact(
        artifact=content,
        metadata=metadata
      )
      related_data_sources.append(
        ds_metadata.id
      )
      ds.data = ds_metadata.id
    return env, related_data_sources

  @classmethod
  def _deserialize(
    cls, env: Env, storage: TupliStorage
  ) -> Env:
    for ds in env.unwrapped.data_sources:
      ds = storage.load_artifact(ds.data)
      ds = ds.decode('utf-8')
      d = io.StringIO(ds)
      df = pd.read_csv(d)
      ds.data = df
    return env

# Instantiate API storage object
tupli_storage = TupliAPIClient()
tupli_storage.set_url(
  "https://company-b-server.com/api"
)
```

```
# Create two users, one for Company B and one for University A
tupli_storage.signup(username="bob_B", password="abc123")
tupli_storage.signup(username="alice_A", password="xyz789")
# Login
tupli_storage.login(username="bob_B", password="abs123")
# Instantiate gymnasium environment
data_paths = [
  "load_data.csv","pv_data.csv","temperature_data.csv"
]
custom_env = PowerSystemEnv(data=data_paths)
# Wrap environment using customized benchmark wrapper
tupli_benchmark = CustomTupliWrapper(
  env=custom_env, storage=tupli_storage
)
# Store and publish the benchmark
tupli_benchmark.store(
  name='EMS_benchmark',
  description="Energy management system control task"
)
tupli_benchmark.publish()
# Load the historical data
historic_episodes = load_historic_data()
# Record and publish the episodes
for eps in historic_episodes:
  eps_item = Episode(
    benchmark_id=tupli_benchmark.id,
    metadata=eps.metadata,
    tuples=eps.tuples
  )
  eps_header = tupli_storage.record(eps_item)
  tupli_storage.publish(eps_header.id)
```

Figure 5: Usage example: Workflow for Company B.

has several parameters that vary depending on the household, most importantly, the load, generation, and outdoor temperature profiles, which are given as CSV files. To create the benchmark that will be used to test the offline RL baseline, Bob chooses data from one of the households that have agreed to the usage. After the gymnasium environment is fully specified, it has to be serialized for storage. Bob has written a subclass of the TupliEnvWrapper that adjusts the methods _serialize() and _deserialize() such that CSV files are stored separately and their reference replaces the data in the benchmark. These functions will be called in the store() functionality of the TupliEnvWrapper. Now, Bob can instantiate and upload a benchmark. He then publishes it to grant access to Alice. To create the dataset for the offline RL training, Bob uses data spanning several households and years. One episode therein corresponds to one day. Bob adds metadata for each episode, such as a household identifier or the month of the year. Each episode is then uploaded with the id of the previously created benchmark.

The second workpackage is completed by University A. As shown in Fig. 6, Alice must first create her own instance of the TupliAPIClient and log in to acquire the necessary rights. Then, she can download the benchmark with the id Bob has given her. She can now download all existing episodes for this benchmark. However, she has the idea of training a seasonal baseline controller and thus adds a filter to retrieve only episodes from the summer months. Alice uses the conservative Q-learning (CQL) implementation from d3rlpy to train the baseline. She converts the downloaded TupliDataset into the MDPDataset format defined by d3rlpy. Finally, she tests the trained controller by running a few episodes on the simulated benchmark. She uses the TupliEnvWrapper to record these test interactions, automatically adding episodes generated by her trained controller to the shared benchmark. This incremental expansion of the dataset with synthetic episodes allows Alice to contribute her policy's behavior back to the collaborative repository, enriching the available data for future research iterations.

For brevity, we only describe potential further steps without providing a code example. Alice could request that Bob continuously add new episodes from the original household as additional data becomes available over time. Then, Alice could retrain her model to assess the effects of increasing datasets and mixing synthetic and real-world data. Furthermore, Bob could create additional benchmarks, one for each household with the corresponding time series data. Alice could use these

```
# Instantiate API storage object
tupli_storage = TupliAPIClient()
tupli_storage.set_url(
  "https://company-b-server.com/api"
)
# Login
tupli_storage.login(
  username="alice_A", password="xyz789"
)
# We assume that this is the id of the
# previously stored benchmark
stored_id = "dl345kn456mlkl230"
# Download benchmark
loaded_tupli_env = CustomTupliWrapper.load(
  storage=tupli_storage,
  benchmark_id=stored_id
)
# Create dataset containing all episodes
# recorded during the summer months
mon = ["June", "July", "August"]
filter_summer = FilterOR(
  filters=[
    FilterEQ(key="month", value=m) for m in mon
  ]
)
filter_benchmark = FilterEQ(
  key='id', value=stored_id
)
dataset_summer = TupliDataset(
  storage=tupli_storage
).with_benchmark_filter(
  filter_benchmark
).with_episode_filter(filter_summer)
dataset_summer.load()
```

```
from d3rlpy.algos import CQLConfig
from d3rlpy.dataset import MDPDataset
# Convert to d3rlpy dataset
obs, act, rew, term, trunc = dataset_summer.convert_to_numpy()
d3rlpy_dataset = MDPDataset(
  observations=obs, actions=act,
  rewards=rew, terminals=term, timeouts=trunc
)
# algorithm for offline training: CQL from d3rlpy
algo = CQLConfig().create(device='cpu')
# train
algo.fit(
  dataset=d3rlpy_dataset, n_steps=10000, n_steps_per_epoch=100
)
# Test trained baseline
# activate recording of episodes
loaded_tupli_env.activate_recording()
# run the environment
obs, info = loaded_tupli_env.reset(seed=42)
for step in range(1000):
  action = np.int64(
    algo.predict(np.expand_dims(obs, axis=0))[0]
  )
  obs, reward, done, truncated, info = loaded_tupli_env.step(
    action
  )
  if done or truncated:
    obs, info = loaded_tupli_env.reset()
```

Figure 6: Usage example: Workflow University A.

benchmarks to finetune her baseline using an online RL algorithm such as SAC. Finally, the trained controllers could be serialized and uploaded as artifacts such that Bob can deploy them in the real households.

PyTupli facilitates this collaboration in multiple ways. Most importantly, it significantly reduces the effort of setting up an infrastructure that enables fast up- and downloads of RL tuples. Furthermore, it automates the conversion of downloaded tuples into the dataset format required by d3rlpy. The filtering capabilities allow University A to hone the quality of the dataset, a crucial aspect in offline RL. Finally, the wrapper for gymnasium environments enables both sides to convert any interaction with the simulation model – for example, during hyperparameter tuning – to be recorded and uploaded as additional tuples.

## 5 CONCLUSION

PyTupli addresses a critical infrastructure gap in collaborations that use offline RL to train controllers for custom tasks. It provides a scalable, containerized solution for creating, storing, and sharing benchmark problems and corresponding tuple datasets. Existing datasets can be filtered at multiple levels, including specific tuples, and converted to the most common input format for offline RL algorithms. While PyTupli significantly improves dataset management for offline RL, its scope has certain limitations. The API storage is based on MongoDB, which may pose scalability constraints for very large deployments. Furthermore, users may need to adjust the output format of datasets to match the input requirements of specific algorithms. Lastly, fine-grained access for user groups is not supported. Despite the current limitations, PyTupli represents the first production-ready collaborative tool in the space of offline RL and, therefore, holds relevance for practitioners from industry and research alike.

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

## A  PREDEFINED ROLES

Table 1: Predefined roles and their associated rights.

| Rights | admin | content_admin | user_admin | group_admin | guest | contributor | member | global_member |
|---|---|---|---|---|---|---|---|---|
| artifact_create | ✓ | ✓ | | | | ✓ | | |
| artifact_delete | ✓ | ✓ | | | | | | |
| artifact_read | ✓ | ✓ | | | ✓ | ✓ | ✓ | ✓ |
| benchmark_create | ✓ | ✓ | | | | ✓ | | |
| benchmark_delete | ✓ | ✓ | | | | | | |
| benchmark_read | ✓ | ✓ | | | ✓ | ✓ | ✓ | ✓ |
| episode_create | ✓ | ✓ | | | | ✓ | | |
| episode_delete | ✓ | ✓ | | | | | | |
| episode_read | ✓ | ✓ | | | ✓ | ✓ | ✓ | ✓ |
| group_create | ✓ | | | ✓ | | | | ✓ |
| group_delete | ✓ | | | ✓ | | | | |
| group_read | ✓ | | | ✓ | | | ✓ | |
| group_update | ✓ | | | ✓ | | | | |
| role_management | ✓ | | | | | | | |
| user_create | ✓ | | ✓ | | | | | |
| user_delete | ✓ | | ✓ | | | | | |
| user_read | ✓ | | ✓ | ✓ | | | ✓ | ✓ |
| user_update | ✓ | | ✓ | | | | | |

**Automatic Memberships**
- Global group: `global_member` + `contributor`
- Private workspace: `user_admin` + `contributor`

## B  API ENDPOINTS

Table 2: PyTupli API endpoints.

| Endpoint | Description | Required Right |
|---|---|---|
| **Artifact Management** | | |
| /artifacts/upload | Upload new artifact file | `artifact_create` |
| /artifacts/list | List accessible artifacts | `artifact_read` |
| /artifacts/download | Download artifact file | `artifact_read` |
| /artifacts/publish | Publish artifact to group | `ownership` and `artifact_create` |
| /artifacts/unpublish | Remove artifact from group | `ownership` or `artifact_delete` |
| /artifacts/delete | Delete artifact permanently | `ownership` or `artifact_delete` (global) |
| **Benchmark Management** | | |
| /benchmarks/create | Create new benchmark | `benchmark_create` |
| /benchmarks/load | Load full benchmark data | `benchmark_read` |
| /benchmarks/list | List accessible benchmarks | `benchmark_read` |
| /benchmarks/publish | Publish benchmark to group | `ownership` and `benchmark_create` |
| /benchmarks/unpublish | Remove benchmark from group | `ownership` or `benchmark_delete` |
| /benchmarks/delete | Delete benchmark permanently | `ownership` or `benchmark_delete` (global) |
| **Episode Management** | | |
| /episodes/record | Record new episode | `episode_create` |
| /episodes/list | List accessible episodes | `episode_read` |
| /episodes/publish | Publish episode to group | `ownership` and `artifact_create` |
| /episodes/unpublish | Remove episode from group | `ownership` or `artifact_delete` |
| /episodes/delete | Delete episode permanently | `ownership` or `episode_delete` (global) |
| **User Management** | | |
| /access/users/create | Create new user | `user_create` |
| /access/users/list | List all users | `user_read` |
| /access/users/delete | Delete user | `ownership` or `user_delete` (global) |
| /access/users/change-password | Change user password | `ownership` or `user_update` (global) |
| /access/users/token | Login and get tokens | - |
| /access/users/refresh-token | Refresh access token | - |
| **Group Management** | | |
| /access/groups/create | Create new group | `group_create` |
| /access/groups/list | List accessible groups | `group_read` |
| /access/groups/read | Get group with members | `group_read` |
| /access/groups/delete | Delete group | `group_delete` |
| /access/groups/add-members | Add members to group | `group_update` |
| /access/groups/remove-members | Remove members from group | `group_update` |
| **Role Management** | | |
| /access/roles/create | Create new role | `role_management` (global) |
| /access/roles/list | List all roles | `role_management` (global) |
| /access/roles/delete | Delete role | `role_management` (global) |

