# OpenReview forum: "PyTupli: A Scalable Infrastructure for Collaborative Offline Reinforcement Learning Projects"
_ICLR.cc/2026/Conference — ICLR 2026 Conference Withdrawn Submission_

### Official Review · Reviewer_2C43 · 2025-10-27

**Soundness:** 3
**Presentation:** 3
**Contribution:** 3
**Rating:** 4
**Confidence:** 4

**Summary:**

The paper introduces PyTupli, a software infrastructure designed to facilitate collaborative offline RL projects. It provides a Python client library and a containerized server backend to manage the entire lifecycle of offline RL assets: creating, storing, sharing, and curating benchmark environments (based on Gymnasium), associated artifacts, and the resulting tuple datasets. The system features role-based access control, a REST API, and advanced filtering capabilities at the benchmark, episode, and tuple levels, aiming to streamline collaboration and improve the reproducibility of offline RL research.

**Strengths:**

- Problem Significance: The paper tackles a real and important problem in applied and collaborative offline RL research: the management of datasets, environments, and artifacts.

- System Design: The architecture is modern, robust, and well-conceived, relying on standard open-source components, which facilitates deployment and maintenance. The inclusion of features like user management, access control, and a production-ready Docker Compose setup is a major strength.

- Usability: The design of the client library with its wrapper, dataset, and storage classes, along with powerful filtering capabilities, appears highly usable and well-integrated with the RL ecosystem (Gymnasium, d3rlpy).

- Clarity: The paper excels in clearly communicating the problem, the proposed solution, and its practical application.

**Weaknesses:**

- No Empirical Evaluation: The primary weakness is the complete lack of quantitative evaluation. The title claims scalability, but no evidence is provided to support this. Benchmarks on ingestion/download speeds, query times on large datasets, or memory/CPU usage under load are necessary to validate such claims.

- Limited Comparison to Alternatives: The paper briefly dismisses tools like Git but fails to compare PyTupli against more relevant data-centric tools like DVC (Data Version Control) or MLOps platforms like MLflow and Weights & Biases. These tools offer overlapping functionality for artifact/dataset versioning and storage. A thorough comparison is needed to clearly delineate PyTupli's unique value proposition for the offline RL community.

- Potential Fragility of Serialization: Relying on a general-purpose serialization library like jsonpickle for complex simulation environments can be problematic. While the authors state this method can be overridden, this offloads a non-trivial engineering challenge to the user. The paper could benefit from discussing best practices or providing more guidance on this for complex use cases.

- Source Code: The authors' usage instructions imply the code is available in a repository, however, that's not the case.

**Questions:**

- The claim of "scalability" is central to the paper's title but remains unsubstantiated. Can you provide any performance benchmarks (e.g., data ingestion/retrieval throughput, query latency as a function of dataset size) to support this claim?

- Could you elaborate on why PyTupli is a superior solution for offline RL projects compared to a combination of existing MLOps/data-versioning tools like DVC or MLflow, which also provide mechanisms for tracking and storing datasets and artifacts alongside code? What specific RL-centric functionalities does PyTupli provide that are critically missing from these more general tools?

- The conclusion states that "fine-grained access for user groups is not supported." This seems to conflict with Section 3.2.1 and Figure 3, which describe a role-based access control system with groups and customizable roles. Could you please clarify this apparent contradiction and specify what level of granularity is missing?

---

### Official Review · Reviewer_xYkB · 2025-11-01

**Soundness:** 2
**Presentation:** 2
**Contribution:** 1
**Rating:** 2
**Confidence:** 4

**Summary:**

The paper proposes a PyTupli library, a Python tool for creating and sharing tuple datasets for
custom Gymnasium environments to be used by offline RL practitioners and researchers. Through containerization, PyTupli enables users to host their own database with a concise API for uploading, downloading, filtering and sharing benchmarks and the corresponding tuple datasets. PyTupli also features user access management. Paper claims that PyTupli fulfils the following functional requirements: benchmark and artifact management, data management, multi-user collaboration and access control, and integration with existing offline RL infrastructure.

**Strengths:**

To avoid sounding overly harsh, I would like to start by noting that the paper is written in a clear and comprehensive manner. I particularly appreciate the motivating stories that help illustrate the motivation and potential workflows. Unfortunately, and with all due respect, this is where the paper’s main strengths seem to end. It is possible that the issue lies more in the presentation (in terms of the narrative and the experiments) rather than in the underlying work itself. However, as a reviewer, I can only evaluate what is presented in the paper in its current form. In the following, I will elaborate on these issues in more detail.

**Weaknesses:**

Firstly, the entire paper consists solely of a dry description of the structure and functioning of the proposed system, completely omitting any experiments or metrics. Since this is purely an engineering contribution and does not offer new scientific knowledge, it must instead demonstrate the practical benefits and in a context relevant to the original motivation in a very clear and understandable way. The value of such work lies not in the quantity and complexity of the work done, but in its ultimate value to users. If it is a complex system with no practical impact, it is significantly worse than an extremely simple but useful one. After all, the user will not know much about the technical details. It is particularly important to demonstrate the advantages over previous solutions, which is completely absent now.

For example:

- numerically demonstrate the effectiveness of your system: what new storage methods do you use to speed up downloads and uploads? As literally storing tuples that duplicate states and double the memory requirements is not optimal.
- What compression methods do you use to save on cloud storage compared to previous solutions for states and images?
- How would you system scale to very large datasets, both for recording and filtering, what are the limitations?
- Do you allow async writing to speed up data collection?
- How much more effective is your solution compared to Minari, RoboDM [1], Lerobot, and HF datasets in general?
- What unique problems your solution solves that others can not? etc.

From the current version of the paper it is not clear. I think RoboDM [1] is a good recent example, and I advise authors to carefully analyse the narrative structure of it. The problem (cost of cloud storage, sampling speed) is clearly demonstrated, and experiments show the effectiveness of the proposed solution.

Second, I don't think the proposed system offers anything new or authors were unable to clearly demonstrate that. Of course, the existence of previous solutions (e.g. Minari) does not mean that one should not try to create something new to solve the same problems. But to do so, one must demonstrate new capabilities. More specifically: most components described in the paper are already provided by Minari.

To my knowledge, in Minary users can already record datasets with the wrapper analogous to the proposed by PyTupli, can serialize and reproduce the exact Gymnasium environments from the recorded datasets, can sample and filter datasets (filtering by tuples is not a significant contribution, as storing large datasets in this way is not efficient anyway), can share them via the cloud (users can specify their custom s3 buckets for storage) and finally, Minary already supported by d3rlpy and torchRL libraries for training. I would be grateful if authors could correct me with specific examples.

This are two major problems with the paper that currently prevent me from endorsing its acceptance. I'm not sure they can be corrected during the rebuttal.

References:

1. Chen, K., Fu, L., Huang, D., Zhang, Y., Chen, L. Y., Huang, H., ... & Goldberg, K. (2025). Robo-DM: Data Management For Large Robot Datasets. *arXiv preprint arXiv:2505.15558*.

**Questions:**

1. Could you clearly state and compare your solution with the Minary? What new capabilities your library unlock?
2. How you store the data? As tuples? Then you should acknowledge the limitations of such approach.
3. Can you describe why current existing solutions (minary + hf datasets) are not enough?

---

### Official Review · Reviewer_otBF · 2025-11-01

**Soundness:** 2
**Presentation:** 2
**Contribution:** 1
**Rating:** 2
**Confidence:** 4

**Summary:**

This paper introduces PyTupli, a Python library designed to facilitate the creation and sharing of tuple-based datasets for custom Gymnasium environments. The tool aims to support offline reinforcement learning (RL) researchers by offering containerized dataset hosting and a straightforward API for uploading, downloading, filtering, and sharing benchmarks and associated datasets. PyTupli also incorporates user management features for collaborative settings. The authors claim the system addresses key functional requirements such as benchmark and artifact management, multi-user collaboration, and integration with existing offline RL workflows.

**Strengths:**

The manuscript is clearly structured and well written, making it easy to follow even for readers outside the immediate area of offline RL tools. The inclusion of motivating examples and user stories helps to ground the technical contribution in practical scenarios. These aspects make the paper accessible and give readers a sense of how the proposed tool might be applied in real research contexts.

**Weaknesses:**

The paper’s most significant limitation is that it remains a purely descriptive system paper without any empirical or comparative evaluation. While the system architecture is explained in detail, the authors do not provide quantitative evidence that PyTupli improves storage efficiency, scalability, or usability over existing tools. For a paper that positions itself as an engineering contribution, the lack of metrics or experimental benchmarks undermines its credibility.

Furthermore, the novelty and differentiation of PyTupli relative to existing libraries - most notably Minari - are not convincingly established. Many of the features highlighted by the authors (environment recording, dataset serialization, and filtering) already exist in Minari, and the paper does not clarify what distinctive benefits PyTupli offers. Without a clear comparison, it is difficult to assess the need for a new tool in this space.

**Questions:**

How does PyTupli differ concretely from Minari and related frameworks? A systematic comparison would greatly strengthen the paper. For instance, does PyTupli introduce new capabilities for user access control, dataset versioning, or distributed collaboration that Minari currently lacks? Are there improvements in ease of deployment or integration with containerized research environments? Presenting a table or case study comparing both systems would help establish PyTupli’s unique value proposition.

How is data storage structured and optimized within PyTupli? Since the paper mentions tuple datasets, it would be important to explain whether the system literally stores full tuples (state, action, reward, next_state) as individual entries. This format can be redundant and inefficient for large datasets, especially if states or images are repeated across tuples. The authors should discuss how PyTupli mitigates these storage inefficiencies - e.g., through compression, deduplication, or asynchronous writing - and how performance scales with dataset size.

---

### Official Review · Reviewer_Dgpb · 2025-11-02

**Soundness:** 2
**Presentation:** 3
**Contribution:** 2
**Rating:** 4
**Confidence:** 4

**Summary:**

This paper discusses the design of PyTupli, a Python package for creating and sharing "tuple datasets" for offline reinforcement learning, where tuples are transitions containing (state, action, next state, reward).

The key idea of PyTupli is to provide both a client-side API and a server for uploading, downloading, and sharing both benchmark results and datasets. The system includes tools for recording interactions from Gymnasium-style environments, serializing them into structured datasets, and storing them alongside environment metadata and artifacts. On the server side, PyTupli offers a FastAPI-based backend with MongoDB storage, user authentication, and fine-grained access control to support collaboration across teams or institutions. The framework also provides dataset filtering and querying utilities, allowing users to select subsets of data based on benchmark attributes, episode metadata, or reward thresholds.

**Strengths:**

Overall, I felt that the PyTupli paper does a good job of articulating its design motivations and potential to standardize how offline RL datasets are shared and benchmarked.

PyTupli recognizes and addresses important problems: offline RL datasets need to be stored, accessed, and filtered. The solution seems clear and reasonable, combining a simple client API with a deployable backend that emphasizes reproducibility and collaboration. The system design is modular: researchers can record tuples locally, filter or subset them through a consistent interface, and later upload them to a shared database without changing code.

The paper includes important details like metadata and access control, which seem critical for usability.

**Weaknesses:**

I am mixed about this paper. On one hand, it's clear that open-source contributions of this flavor have been crucial in enabling the last decade of progress in AI research. Venues like ICLR also have a track record of publishing technical reports on software; for example PyTorch at NeurIPS 2019 or Einops at ICLR 2022.

I do, on the other hand, have some concerns about PyTupli. The main thing is that it's difficult without either (i) existing adoption or (ii) more concrete evaluation to assess how well PyTupli addresses the needs of the research community it is designed for.

While data management is uncontroversially critical in the field, I would expect data management needs to vary significantly between teams and collaborations. It's not clear to me that a standardized, offline RL-specific approach for managing datasets is the correct way forward. (compared to building off of more generic serialization and storage tools)

PyTupli is built on a MongoDB database, with a custom username/password system for managing access. I'm concerned about some practical ramifications of this:
- Cloud storage solutions like S3 or GCS coupled with some indexing mechanism are the existing standard for handling large datasets; this seems easier to scale and probably more cost-efficient, particularly when taking into account cold vs hot storage rates.
- All data that's accessed needs to come through the same containerized MongoDB instance. Throughput issues seem likely when there are many users who all want to train on the same data.
- "Rolling your own auth" is often considered poor practice. This may limit adoption in institutions with stricter security policies.

As a result of the notes above, I'm currently rating the paper a weak accept. I'm happy to revisit based on author responses and discussion with other reviewers.

**Questions:**

For hosting/sharing large datasets, a key driving factor is often cost. Are you able to estimate the cost of hosting data via PyTupli's existing architecture as opposed to, as an example, on S3?

Is there existing adoption of PyTupli, or examples of existing applications? This would be useful context for evaluating the utility of the project and relevance to the ICLR community.

A key motivation for the PyTupli framework is scalability; the title of the work itself describes PyTupli as "scalable infrastructure". Can you define what kind of scalability you're referring to?

---

### Note · Authors · 2025-12-10

I have read and agree with the venue's withdrawal policy on behalf of myself and my co-authors.